# Study protocol for a multi-center stepped-wedge cluster randomized trial to explore the usability and outcomes among young people living with HIV in Kiambu and Kirinyaga counties of Kenya, using an online health portal

Eric Nturibi[1]◉*, Jared Mecha[1]◉, Elizabeth Kubo[1]◉, Albert Orwa[1]◉, Florence Kaara[1‡], Faith Musau[1‡], Christine Wamuyu[1‡], Justus Kilonzi[2‡], Randeep Gill[3]◉, Sanne Roels[3‡]

1 Department of Clinical Medicine and Therapeutics, University of Nairobi, Nairobi, Kenya, 2 Global Health Policy, Savannah Global Health Institute, Nairobi, Kenya, 3 Johnson & Johnson Global Public Health, Janssen Pharmaceutica NV, Beerse, Belgium

◉ These authors contributed equally to this work.
‡ FK, FM, CW, JK, and SR also contributed equally to this work.
* ericmugambi@gmail.com

**Data Availability Statement:** This is a protocol and data will be fully available on analysis.

## Abstract

While the incidence of Human Immunodeficiency Virus (HIV) infection is decreasing in most age groups worldwide, it is rising among adolescents and young adults, who also face a higher rate of HIV-related deaths. This tech-savvy demographic may benefit from an online patient portal designed to enhance patient activation—empowering them to manage their health independently. However, the effectiveness of such digital health interventions on young HIV patients in Kenya remains uncertain. We will conduct a 12-month stepped wedge cluster randomized trial involving 15-24-year-old HIV patients with smartphone access. The primary outcome will be patient activation, with secondary outcomes including self-reported adherence, social engagement and viral suppression. We will also evaluate the portal's functionality, usability, fidelity, and costs. Participants will be recruited from 47 antiretroviral treatment (ART) sites with electronic medical records (EMR), forming 16 clusters of 30 participants each. Clusters will be randomized into three sequences for intervention every three months. Baseline measurements (patient activation, adherence, social engagement and viral suppression) will be collected over two weeks, followed by checks at 3, 6, and 12 months. Data will be analyzed using generalized linear mixed models and adjusted for cluster effects and potential confounders. Results will be disseminated through stakeholder forums, scientific conferences, peer-reviewed publications, and the media.

**Funding:** This study was supported by the Patient Health Portal Project Grant (Number 66671241) awarded to the University of Nairobi, where EN, JM, EK, AO, FK, FM and CW are employees (www. uonbi.ac.ke). JM served as the overall institutional lead on the side of the University of Nairobi. JK is employed by Savannah Global Health Institute, the organization contracted to develop the portal. The grant was provided by the Johnson & Johnson Foundation, Scotland (www.jjfscotland.org). RG and SR are affiliated with Janssen Pharmaceuticals NV, a subsidiary of Johnson & Johnson. RG contributed to the study design and conceptual framework, and SR contributed to sample size estimation. The funders had no role in data collection and analysis, decision to publish, or preparation of the manuscript.

**Competing interests:** Randeep Gill and Sanne Roles are employees of Janssen Pharmaceuticals NV, a subsidiary of Johnson & Johnson, which funded the study. Both contributed to the study's design and conceptual framework. Their employment may be viewed as a competing interest.

## Author summary

HIV infections and deaths among teenagers and young adults are rising, with the situation being particularly concerning in Kenya. This age group's familiarity with technology offers a potential avenue to address this issue through an online health portal, which could help them better manage their health. However, the effectiveness of such an approach is not yet clear. To investigate this, we are conducting a year-long study involving HIV-positive individuals aged 15 to 24 who have access to smartphones. The study's primary goal is to evaluate whether the portal improves patient activation, medication adherence, HIV control, and social engagement. Participants from 47 HIV treatment centers will engage with the portal in staggered groups every three months. Statistical models will assess its impact on patient outcomes, while usability, functionality, and operational costs will also be analyzed. Findings will be shared with the community, presented at conferences, and published in scientific journals and mainstream media.

## Introduction

Of the 37.7 million people living with the Human Immunodeficiency Virus (PLHIV) globally in 2020, 25.3 million resided in sub-Saharan Africa (SSA), where 460,000 of the 680,000 global acquired immune deficiency syndrome (AIDS) related deaths (ARD) also occurred [1]. While new infections are decreasing in other age groups, they continue to rise among adolescents and young people (AYP). In 2020 alone, 400,000 new cases were reported, with 150,000 occurring in individuals aged 10 to 19—an age group also reported to be disproportionately affected by ARD [2,3]. In Kenya, there were an estimated 1.4 million PLHIV, including 160,000 young people aged 15–24 by the end of 2020. Of the 33,000 new infections recorded in the same year, approximately 35% were among this younger age group [4]. As of January 2021, Kenya had nearly 60 million mobile phone connections, resulting in a mobile penetration rate of 108.9%. The country's internet penetration stands at 40%, representing over 21 million people, about half of whom also used social media [5]. According to the International Telecommunication Union, young people in low- and middle-income countries (LMICs) are nearly three times more likely to be using the internet compared to the general population [6]. Despite this technological readiness, a recent review highlighted the lack of published interventions targeting the over 4 million young people aged 15–24 living with HIV globally, despite being identified as a "critical group experiencing multiple barriers to engagement in care." [7]. Patient health portals offer a promising solution for this vulnerable population, who may be more likely to adopt digital health solutions compared to other age groups. To address this need, we developed the myCareHub digital health platform with input from the intended users, incorporating functionalities and mechanisms critical to patient activation (Fig 1) [8]. Grounded in the COM-B framework for behavior change, which posits that behavior is influenced by three key factors—capability, opportunity, and motivation—the platform is designed to address each of these elements to promote effective self-management of health (Fig 2) [9].

The platform will be accessible via password-protected mobile phones to ensure data security and confidentiality, offering a range of services to both healthcare providers (HCPs) and patients (Fig 3).

This study will evaluate whether myCareHub can improve patient activation among AYPs, leading to greater engagement in HIV care and treatment. In alignment with the World Health Organization (WHO) guidelines for mobile health evidence reporting and assessment

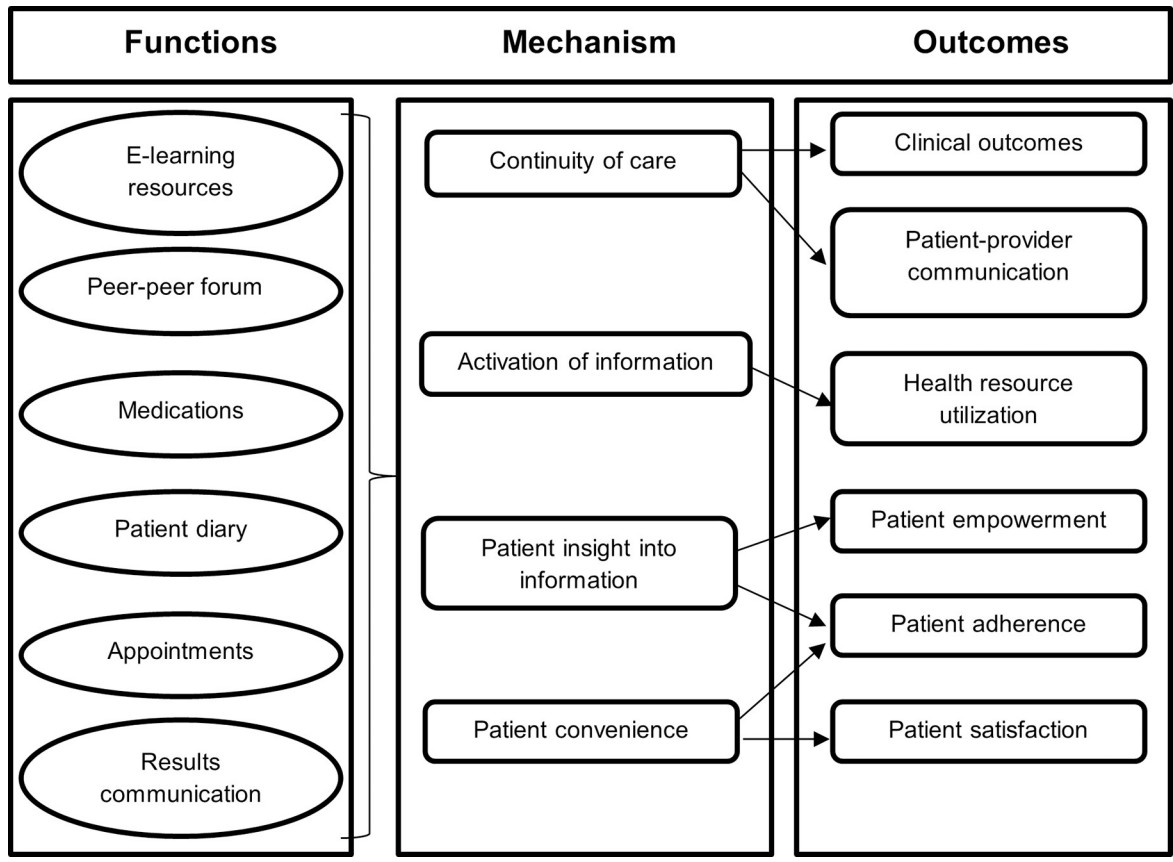

**Fig 1. Mechanisms of patient activation—adapted from Otte-Trojel, de Bont [8].**

(mERA), the study will also assess the quality, fidelity, and coverage of the platform, focusing on key program indicators often overlooked in published studies (Fig 4) [10].

The results of this study will inform HIV programming for young people and other vulnerable populations, contribute to digital health policy, and play a key role in advancing the global goal of eliminating AIDS as a public health threat by 2030.

## Materials and methods

### Setting

The study will take place as part of a large antiretroviral therapy (ART) program in partnership with the Kiambu and Kirinyaga county health departments. The program currently supports HIV-related prevention, care, and treatment activities at 78 public healthcare facilities, with 47 of these facilities equipped with electronic medical records (EMR) systems.

For this study, the 47 EMR-enabled facilities will be clustered based on geographical proximity, roughly coinciding with sub-county territorial boundaries. At study inception, the program provided ART to 38,079 PLHIV, including 3,459 (9%) AYP aged 15–24. Around 89% of the target population is enrolled at the 47 EMR sites, which will be the primary focus for participant recruitment.

### Study design

This is a multicenter stepped wedge cluster randomized trial designed to evaluate the impact of the myCareHub patient portal on improving patient activation and clinical outcomes

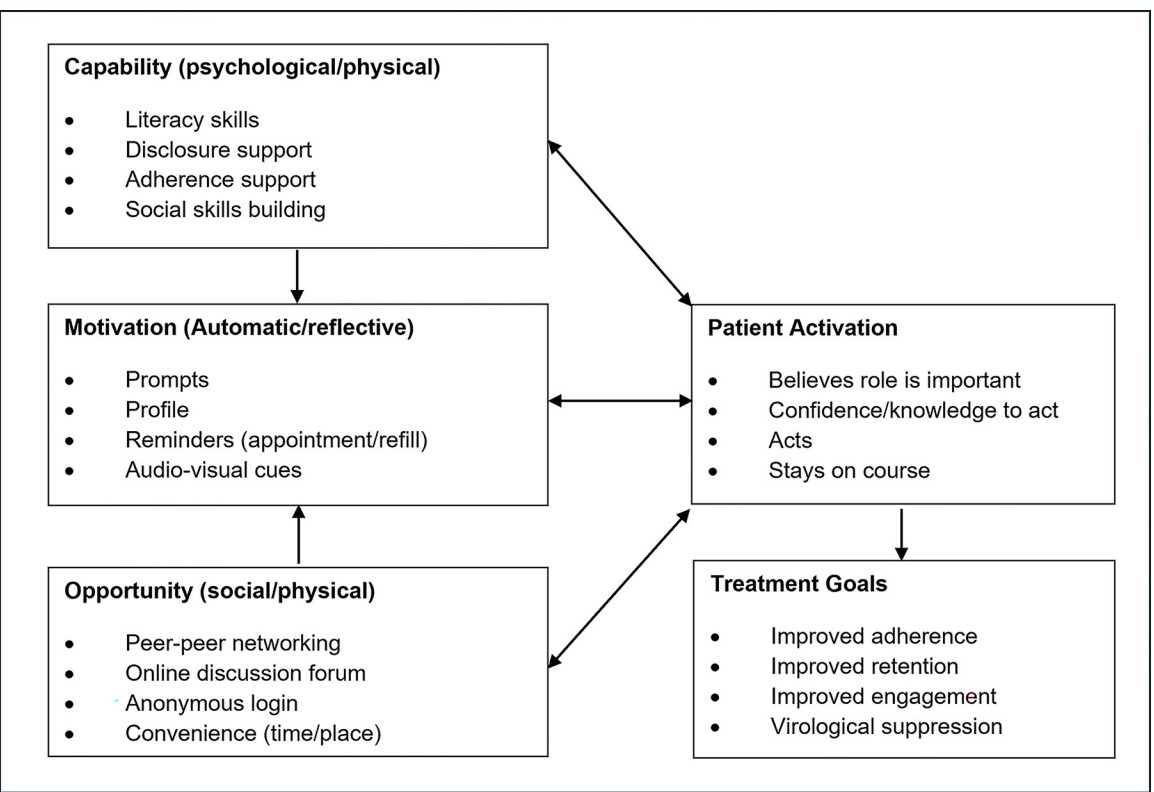

**Fig 2. COM-B theoretical framework for behaviour change.**

among HIV- positive AYP aged 15–24 years. Participants will be recruited from 47 EMR-enabled ART clinics (Table 1).

## Justification for the stepped wedge design

The stepped wedge design was chosen to mitigate contamination bias. In a facility-based setting, randomizing individuals to control or intervention groups within the same facility would result in unintentional exposure of the control group to the intervention. The staggered implementation across clusters also enables continuous refinement of the myCareHub platform based on feedback from earlier clusters, thus improving the intervention over time.

A minimum sample of 480 AYP will be recruited into 16 clusters using a stepwise design, with three-month intervals between successive steps (Table 2).

To minimize heterogeneity between clusters, stratification was applied based on size (small, medium, large), sex (male, female), and viral load suppression (VLS). This stratification created comparable blocks of clusters prior to randomization, ensuring balance across the intervention groups (Table 3). Clusters were then randomly assigned to one of three sequences for the time of crossover from control to intervention using a computer-generated random number list (Table 4). Each cluster will begin in the control period and transition to the intervention at staggered intervals, with clusters randomized to groups of five or six. Each intervention sequence will last three months, with a two-week interval between sequences to allow for baseline measurements and smooth implementation of the intervention. The third sequence will extend for six months to allow for additional measurements (Table 2).

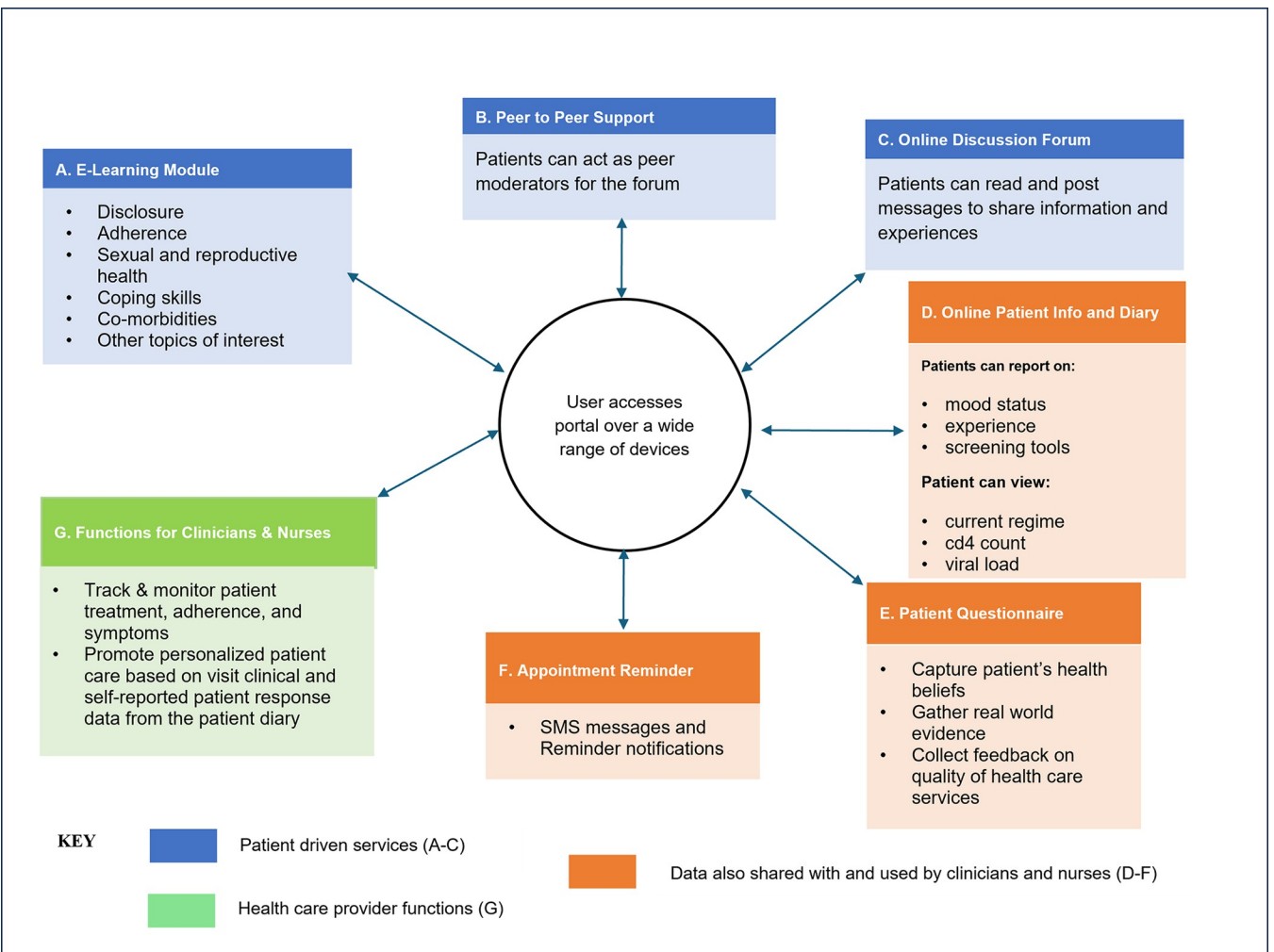

**Fig 3. Diagrammatic representation of myCareHub and functions available to use.**

## Study population

In this trial, the intervention will target clusters, where a cluster is defined as a group of EMR—capable facilities within a specific geographical proximity. Outcome parameters will be assessed at the patient level.

Two recruitment strategies will be used to enroll participants:

1. Line listing—a central EMR database will provide a line list of all AYP in the participating EMR sites. Random number sequences will be used to select eligible participants at each facility, with allowances for refusals.

2. Weekly appointment diaries—a weekly diary will be used to list AYP scheduled for clinic visits in the coming week. Eligible participants will be randomly selected using a random number sequence.

Selected participants will be invited to an open-day activity for recruitment, where informed consent will be obtained. These recruitment strategies will continue until the required sample size is reached.

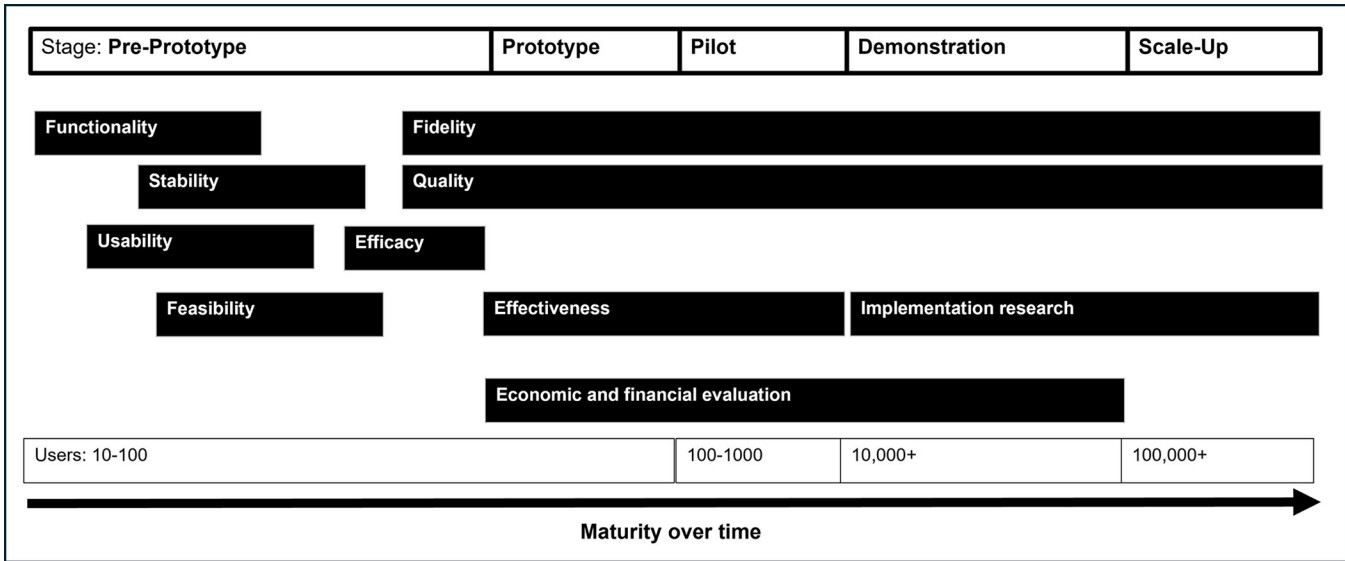

**Fig 4. Life cycle of a digital intervention.**

Measurements will be taken on the same cohort at the end of each intervention step as clusters move from the control to active intervention phases (Table 2).

## Intervention

The myCareHub portal will be available on both the Google Play Store and iOS App Store. Developed by the study team with significant input from end users, it is designed to facilitate behavior change using the COM-B framework (Figs 2 and 3). The portal comprises two mobile-based applications: a patient portal and an HCP portal, offering a range of features critical to supporting patients and providers in HIV care.

**Table 1. PICOT framework.**

| | |
|---|---|
| Population | Adolescents and young persons aged 15–24 years, who are living with HIV and are receiving care at an EMR-enabled Anti-Retroviral Treatment (ART) site. All participants will have ready access to a smartphone, which is necessary for using the myCareHub portal. |
| Intervention | Implementation and testing of the myCareHub patient health portal. This portal is designed with several functionalities that support behavior change according to the COM-B framework. These include e-learning resources, an online patient diary, peer-to-peer networking, patient-reported outcomes, and personalized care triggered by interaction with healthcare providers (HCPs). |
| Comparator | Standard of care patient engagement strategies, which involve traditional in-person clinic visits, standard ART support, and typical adherence and social engagement practices without the use of the myCareHub digital health portal. |
| Outcome | The primary outcome of the study is patient activation, which will be measured using the Patient activation measure (PAM). Secondary outcomes include viral load suppression, social engagement (assessed using the Social support questionnaire [SSQ6]), self-reported ART adherence, and portal usage metrics (e.g., login frequency, time spent, interaction with portal features). |
| Time | The study will span a total of 15 months. Initially, there will be a control period of 3 months, during which clusters will receive standard care. Intervention sequences will be rolled out every 3 months, with measurements taken at baseline and every three months for each cluster. After all clusters have received the intervention, a 6-month observation period will follow. |

**Table 2. Study design—cluster randomized trial utilizing a stepped wedge design with 16 clusters and 3 sequences.**

| | T0 | T1 | T2 | T3 | T4 |
|---|---|---|---|---|---|
| CLUSTERS (11–16) | CONTROL | CONTROL | CONTROL | INTERVENTION | INTERVENTION |
| CLUSTERS (6–10) | CONTROL | CONTROL | INTERVENTION | INTERVENTION | INTERVENTION |
| CLUSTERS (1–5) | CONTROL | INTERVENTION | INTERVENTION | INTERVENTION | INTERVENTION |
| Survey content | T0: Baseline survey: Demographics, clinical enquiry, baseline PAM, adherence, VLS, social engagement scores on SSQ6, cluster level retention, cluster VLS. | T1 survey: 3 Mo PAM, adherence, VLS, portal measures: TB ICF, PHQ-9, CRAFFT, SGBV screening, Contraceptive use intention. Feedback on portal, cluster level retention, cluster VLS. | T2 survey: 3, 6 Mo PAM, adherence, VLS, retention, portal measures: TB ICF, PHQ-9, CRAFFT, SGBV screening, Contraceptive use intention. Feedback on portal, cluster level retention, cluster VLS. | T3 survey: 3, 6, 9 Mo PAM, adherence, VLS, retention, portal measures: TB ICF, PHQ-9, CRAFFT, SGBV screening, Contraceptive use intention, feedback on portal, cluster level retention, cluster VLS. | T4 survey: 6,9 & 12 Mo PAM, adherence, VLS, MAUQ acceptability score, portal measures: TB ICF, PHQ-9, CRAFFT, SGBV screening, Contraceptive use intention, feedback on portal, cluster level retention, cluster VLS. |
| TIME | T0 | T1 Mo 1–3 | T2 Mo 4–6 | T3 Mo 7–9 | T4 Mo 10–12 |
| | 2 weeks | 2 weeks | 2 weeks | 2 weeks | 2 weeks |

**Abbreviations**: TO–baseline, T1 –time period 1, T2 –time period 2, T3 –time period 3, T4 –time period 4. Mo–month. PAM–patient activation measure. VLS–viral load suppression. SSQ6- social support questionnaire short form. MAUQ–mHealth App usability questionnaire. TB ICF–tuberculosis intensified case finding. PHQ-9 –patient health questionnaire– 9 items, SGBV –sexual and gender–based violence, CRAFFT–car, relax, alone, forget, friends, trouble. **NB**: The third sequence is extended to ensure that the final stratum completes a minimum of six months, which is the recommended duration for adequately assessing patient activation.

**Table 3. Sub-counties by key attributes.**

| Sub-county | Number of AYP [b] | Size strata | VLS [a] 15–19 | VLS 20–24 | Cohort retention | Males (%) |
|---|---|---|---|---|---|---|
| Thika town | 632 | Large | 80% | 90% | 89% | 41% |
| Ruiru | 330 | | 84% | 89% | 68% | 25% |
| Kiambu town | 205 | Medium | 86% | 89% | 100% | 32% |
| Kiambu Central | 176 | | 86% | 90% | 84% | 55% |
| Gatundu South | 166 | | 61% | 90% | 50% | 40% |
| Juja | 146 | | 91% | 82% | 67% | 35% |
| Kiambaa | 123 | | 87% | 91% | 100% | 27% |
| Kirinyaga South | 116 | | 91% | 87% | 88% | 53% |
| Githunguri | 113 | | 85% | 89% | 100% | 27% |
| Kabete | 110 | | 94% | 94% | 100% | 23% |
| Gatundu North | 107 | | 92% | 89% | 100% | 40% |
| Limuru | 103 | | 90% | 100% | 100% | 35% |
| Kirinyaga West | 88 | Small | 81% | 92% | 74% | 37% |
| Kikuyu | 73 | | 100% | 90% | 100% | 16% |
| Kirinyaga East | 58 | | 100% | 86% | 87% | 57% |
| Lari | 51 | | 100% | 91% | 100% | 23% |
| Kirinyaga North | 50 | | 60% | 68% | 88% | 48% |

[a] viral load suppression

[b] adolescents and young persons

**Table 4. Randomization of clusters into sequences.**

| Cluster | 15–19 years old | 20–24 years old | Total AYP [a] | Strata Size | Random number | Sequence | No. of AYP (n = 480) |
|---|---|---|---|---|---|---|---|
| Juja | 48 | 98 | 146 | Medium | 0.91 | 1 | 186 |
| Kiambaa | 30 | 80 | 110 | Medium | 0.84 | 1 | |
| Kirinyaga Central | 76 | 83 | 159 | Medium | 0.64 | 1 | |
| Kabete | 19 | 75 | 94 | Small | 0.97 | 1 | |
| Kirinyaga South | 27 | 48 | 75 | Small | 0.83 | 1 | |
| Ruiru | 77 | 253 | 330 | Large | 0.73 | 1 | |
| Gatundu South | 75 | 91 | 166 | Medium | 0.56 | 2 | 170 |
| Gatundu North | 52 | 55 | 107 | Medium | 0.55 | 2 | |
| Limuru | 41 | 62 | 103 | Medium | 0.54 | 2 | |
| Lari | 12 | 23 | 35 | Small | 0.70 | 2 | |
| Kirinyaga East | 29 | 26 | 55 | Small | 0.57 | 2 | |
| Thika Level 5 | 172 | 196 | 368 | Large | 0.23 | 2 | |
| Githunguri | 45 | 56 | 101 | Medium | 0.46 | 3 | 124 |
| Thika Town | 102 | 135 | 237 | Medium | 0.38 | 3 | |
| Kiambu Township | 87 | 112 | 199 | Medium | 0.06 | 3 | |
| Kirinyaga West | 29 | 42 | 71 | Small | 0.51 | 3 | |

[a] adolescents and young people

### Patient portal

Key features available to patients include:

- E-learning resources—text, audio, and video content focusing on HIV treatment literacy, disclosure support, adherence support, sexual and reproductive health, and coping skills.

- Online communities—peer-to-peer networking and learning platforms to foster interaction and support among patients.

- Patient diary—a tool for self-monitoring, allowing patients to view their medical data from the facility's EMR, record their experiences, and share relevant information with HCPs.

- Patient-reported outcomes—web forms and survey tools that capture patient-reported outcomes, triggering service requests in the HCP portal, ensuring timely and personalized care.

- Notifications—in-app and short message service (SMS) notifications for appointments and other reminders.

- Appointment rescheduling—this feature, to be developed in the second phase, will allow patients to reschedule appointments.

### Healthcare provider portal

The health care provider (HCP) portal will provide healthcare providers with tools to:

- Track and monitor patient treatment, adherence, and symptoms.

- Offer personalized care based on data from the patient's diary.

- Access patient data—the portal is synchronized with the facility's EMR, allowing HCPs to access longitudinal patient visit data, lab results, and other medical information.

To ensure optimal use, quarterly open forums will be conducted with both HCPs and patients. These sessions will facilitate access to all available features and capture qualitative end-user insights for further improvements.

### Outcome measures

Outcome parameters will be measured at the individual level at pre-specified time points: baseline, every 3 months, and end-line (Table 5). These measurements will be captured using digitized case report forms (CRFs) to ensure accuracy and consistency in data collection.

### Study tools

**Patient activation measure.** The Patient Activation Measure (PAM) is a tool designed to assess an individual's knowledge, skills, and confidence in managing their own health and healthcare. The measure evaluates patients across four levels of activation. At Level 1, patients may not yet grasp the importance of their role in managing their own health. Level 2 marks the beginning of confidence and knowledge development, though patients might still struggle with maintaining consistent self-management behaviors. By Level 3, patients have gained key information and are taking action, although they may still need guidance to improve or sustain certain health behaviors. Finally, at Level 4, patients are proactive, able to maintain lifestyle changes, and manage their health independently with minimal support. The Patient Activation Measure (PAM) is typically scored on a 100-point scale, where higher scores indicate greater activation [11].

**Table 5. Summary of outcome variables and analysis measures.**

| Variable | Definition | Units | Source | Measurement Level | Measurement Period | Comparison Groups | Analysis |
|---|---|---|---|---|---|---|---|
| PAM | Patient activation measure | 100-point scale | PAM-13 | Individual | Baseline, 3, 6 and 9 months | Intervention versus Control | Mean change in PAM |
| Self-reported medication adherence | Estimate of adherence on 0–100% scale | Proportion | Visual analogue scale and adherence questionnaire | Individual | Baseline, 3, 6 and 9 months | Intervention versus Control | % adherence |
| Viral load suppression (VLS) | Proportion with viral load below detection limit | Proportion | Electronic medical record | Individual | Baseline, 3, 6 and 9 months | Intervention versus Control | % VLS |
| Cluster retention | Active in care at the end of reporting period/active at the beginning of the period*100 | Proportion | Program data | Cluster | Baseline, 3, 6 and 9 months | Intervention versus Control | % retention |
| Cluster VLS | Percent viral load suppression within cluster | Proportion | Program data | Cluster | Baseline, 3, 6 and 9 months | Intervention versus Control | % VLS |
| Mobile health application usability score | Numerical score | Mean score | mHealth App Usability Questionnaire (MAUQ) | Individual | End-point | Intervention | Mean scores |
| Mobile health service acceptability score | Likert-like scale and text | Sub-section scores | Custom acceptability tool | Individual | End-point | Intervention | Mean scores |
| Social support | Six questions on social supporters | Number of supporters and ratings of support | Social support questionnaire-6 item (SSQ-6) | Individual | Baseline | Intervention | Baseline SSQ-6 scores |
| Tuberculosis (TB) intensified case finding tool (ICF) | 4 questions each scoring 0 or 1 | 0–4 | Portal TB ICF tool, Paper-based TB ICF tool, Program reports | Individual | Continuous | Intervention versus Usual care | % cases diagnosed |
| | | | | | | | Median time to TB diagnosis |
| Screening for depression | 9 questions each scored 0–3 for total possible score of 27 | 0–27 | Portal patient health questionnaire (PHQ-9) | Individual | Continuous | Intervention versus Usual care | % cases diagnosed |
| Contraceptive use intention screening | 4 questions | Yes/No | Portal screening tool | Individual | Continuous | Intervention versus Usual care | % users accessing contraceptives |
| Sexual and Gender Based Violence (SGBV) screening | 4 screening questions | Yes/No | Portal SGBV screening tool | Individual | Continuous | Intervention versus Usual care | % survivors successfully identified |
| Alcohol and substance use disorder screening | 6 screening questions | Yes/No, with score>2 indicating problem | Portal incorporated CRAFFT tool (Car, relax, alone, forget, friends, trouble) | Individual | Continuous | Intervention versus Usual care | % with alcohol use disorder identified |
| Online community engagement | User online activity and interactions | Usage logs, time spent on portal, feature usage | Analytical dashboard | Individual | Continuous | Intervention | Online engagement |

**Antiretroviral therapy adherence questionnaire.** To assess medication adherence in participants, we will use a customized ART Adherence Questionnaire utilizing a visual analogue scale (VAS) to estimate self-reported adherence over the past 30 days. Similar VAS measures have been well-validated in previous studies, demonstrating their reliability and simplicity for capturing self-reported adherence, particularly in diverse patient populations [12–14]. In addition to the VAS, our tool incorporates questions about the timing of missed doses and the reasons for non-adherence such as being away from home, forgetting, or concerns about side

effects. This adherence measure enables both a visual and quantitative reflection of participants' adherence behaviors, facilitating tracking of adherence patterns across the study. This tool will be administered at 3, 6 and 9 months.

**Viral load suppression and cluster retention.** A customized data collection form will be used to extract individual viral load suppression status, cluster retention and cluster viral load suppression data from program reports and facility EMRs.

**Evaluating usability.** At the end of the intervention period, we will administer the Mobile App Usability Questionnaire (MAUQ) to all users (HCPs and patients). The tool has shown high reliability in prior studies, with Cronbach's alpha coefficients consistently exceeding 0.80, reflecting strong internal consistency [15]. We have obtained permission to use the MAUQ and will follow standard protocols for its administration and scoring to ensure reliable results in our study.

**Acceptability survey.** The acceptability survey was designed specifically for this study, drawing on validated frameworks for measuring user acceptance and satisfaction [16]. Pilot testing was conducted to refine the survey and ensure its reliability before full-scale administration.

**Social engagement score.** The SSQ6 (Social Support Questionnaire—Short Form) is a well-established tool used to assess perceived social support availability. This tool, which is freely available for research use, measures the number of available supports and rates the participant's satisfaction with that support [17]. We will track user engagement in online communities and compare it with baseline social engagement scores from the SSQ6. This will involve collecting quantitative data on user interactions and activity levels.

**Focus group discussions.** Focus group discussions (FGDs) will be structured to gather qualitative insights from users about their experience with the portal. Discussion guides have been developed based on key usability and satisfaction themes. We will use trained facilitators to ensure consistency and depth in the discussions. Insights from FGDs will be analyzed thematically to identify common issues and suggestions for improvement. This feedback will be used to inform iterative design enhancements to the portal.

**Portal-based measures.** During the study, technical data related to functionality, stability, fidelity, and quality of the myCareHub platform will be continuously collected. The platform will automatically track key metrics, such as user logins, accessed functions, and time spent on various functional components. This usage data will be continuously analyzed to identify patterns and areas for improvement. Descriptive statistics and usage trends will be reported to highlight key findings related to user behaviour and feature performance.

*Functionality*. We will assess whether myCareHub operates as intended by collecting data through the user experience survey, conducting FDGs (qualitative), and using the Usability section of the MAUQ for both patients and HCPs.

*Usability*. We will evaluate the ease of use of myCareHub by using the Functionality section of the MAUQ tool.

*Stability*. The stability of myCareHub will be measured through system reports on server downtime, server operation capacity, network connectivity, SMS/notification failure rates, and using the Stability section of the MAUQ tool.

*Quality*. To assess the quality of myCareHub, we will measure the average time spent on the portal, frequency of use, and functionalities accessed. Qualitative aspects of portal use will also be examined using the MAUQ tool. Additionally, a mobile health service patient acceptability questionnaire will be administered.

*Fidelity*. We will track whether field implementation alters the functionality and stability of the system by reviewing functionality reports, stability reports, and instances of forgotten passwords or incorrect intervention delivery by HCPs.

*Cost*. A comprehensive cost analysis will be undertaken to estimate the total cost of setting up myCareHub. This will include operational costs, such as server expenses and staff training.

*Additional screening tools*. myCareHub users will have access to various screening tools on their health diary, including the tuberculosis (TB) case finding tool, patient health questionnaire– 9 (depression screening), gender-based violence screening, contraceptive need screening, and screening for alcohol and substance use disorders. These tools are free to use and are already integrated as part of the standard of care within the ART program. Once these tools are completed, HCPs will receive an in-app notification indicating a pending task, with an automated status response sent back to the user. Upon task completion at the clinic, the HCP will mark the task as fulfilled, triggering a second notification to the user. myCareHub will automatically track this process cycle, and at the end of the study, the proportion of services fulfilled through the portal will be summarized and compared with the total services offered to the same age group. The median time to diagnosis of TB through the portal will be calculated and compared to usual care.

## Statistical methods

Statistical analysis of the primary and secondary endpoints will be conducted in either R or SAS using generalized linear mixed models (GLMMs), which allow for incorporation of correlations due to repeated measures, cluster effects, and potential time effects.

**Variable selection.** Variables such as age, sex, occupation, duration on ART, presence of a treatment buddy, and last viral load will be included based on their relevance to the study objectives and previous literature. The primary outcome will be the change in patient activation measure (PAM) at 6 and 9 months. Changes in PAM score will be analyzed with respect to intervention and control groups, stratified by age category (15–19 vs. 20–24), sex (male vs. female), and duration on ART (<12 months vs. >12 months).

**Interaction terms.** Interaction terms will be included to explore potential interactions between key variables, such as age and sex, or duration on ART and the intervention group. These interactions help to understand whether the effect of the intervention varies across different subgroups.

**Handling of potential confounding variables and assumptions.** Confounding variables identified through exploratory analyses, such as sociodemographic factors and baseline clinical characteristics, will be included in the model to adjust for their effects on PAM scores. Model diagnostics will be performed to ensure appropriate model fit, including checks for residuals and potential overdispersion. Random effects will account for individual variations and cluster effects. The assumptions of GLMMs, including the normality of random effects and the appropriate link function, will be tested using diagnostic plots and statistical tests.

Sensitivity analyses, including the use of causal inference methods, will be conducted to assess the effect of differential patient and cluster characteristics over time, and attrition on the comparison between intervention periods. These analyses will also evaluate the robustness of findings, including the handling of missing data, such as treating PAM scores of 0 and 100 as missing values.

**Statistical analysis of secondary objectives.** *Self-reported medication adherence*. Self-reported adherence scores, collected at baseline, 3, 6, and 9 months, will be analyzed using GLMMs. These models will account for correlations between repeated measures across the different time points and clustering effects within facilities. The percentage of adherence in the intervention versus control groups will be compared, with adjustments made for potential confounders such as age, sex, and ART duration.

*Viral load suppression*. The proportion of patients achieving viral load suppression (VLS) at baseline, 3, 6, and 9 months will be modeled using logistic regression, adjusted for cluster effects. A comparison between the intervention and control groups will be made, considering time effects and covariates. The percentage of participants in each group who achieve viral suppression will be reported, and changes over time will be assessed.

*Social engagement*. Social engagement scores (SES) from the SSQ6 will be compared across time points (baseline, 3, 6, 9 months) using mixed-effects models. Participation in online communities will be tracked and compared to baseline scores to assess whether the intervention effectively improves social engagement. Differences in social engagement over time will be explored, with subgroup analyses based on age, sex, and baseline social support.

*Portal use and engagement metrics*. The frequency of portal logins, time spent on the portal, and functionalities accessed will be analyzed using descriptive statistics. Regression models will be used to explore relationships between portal use and adherence, activation, and viral suppression outcomes. Portal use will be summarized as engagement metrics (e.g., total logins, average time spent) and correlated with clinical outcomes, including adherence and viral suppression.

*Usability and acceptability scores*. Usability scores from the MAUQ and the custom-designed acceptability survey will be analyzed using descriptive statistics and compared across subgroups (age, sex, ART duration). Generalized linear models will be used to explore the association between usability scores and primary or secondary outcomes. Usability and acceptability will be assessed and reported, with particular attention to how these factors influence engagement with the platform.

**Statistical analysis for portal-based measures.** *Depression screening*. The scores from the patient health questionnaire—9, which range from 0 to 27, will be analyzed using generalized estimating equations (GEE) to account for repeated measures and cluster effects. The proportion of participants identified with mild, moderate, and severe depression will be compared over time and by intervention group. Logistic regression will assess the odds of timely referral for those flagged through the screening. The proportion of participants with depression identified and referred for further evaluation, and percentage of depression cases identified through the portal compared to standard care.

*Contraceptive use screening*. Responses to the contraceptive needs assessment will be analyzed descriptively with logistic regression models exploring factors associated with successful referrals for contraceptive services. The proportion of participants screened and successfully referred for contraceptive use will be tracked and compared between portal-based and standard care. Proportion of contraceptive requests from the portal and successful referrals, compared to total requests made through both portal and standard care.

*Tuberculosis screening*. Screening results for TB will be analyzed using logistic regression to compare referral rates between those screened via the portal and those using standard care. Time to referral following a positive TB screen will be analyzed using Cox proportional hazards models. The proportion of TB cases identified and referred through the portal and percentage of total TB cases identified overall during the study period.

*Alcohol and substance use screening*. Screening data from the CRAFFT (car, relax, alcohol, forget, friends, trouble) tool will be analyzed using GLMMs to assess differences in substance use disorder identification rates between portal and non-portal groups. The proportion of positive screens and subsequent referrals to intervention services will also be compared across the study periods. Percentage of alcohol and substance use cases identified through the portal and successful referrals for follow-up services will be compared to usual care.

*Time to service fulfilment*. Additionally, the time to service provision and HCP responsiveness to screening-related service requests will be compared between portal users and standard

care groups using survival analysis techniques. These analyses will help determine the effectiveness and efficiency of integrating digital screening tools within the myCareHub platform in comparison to traditional care pathways.

*Qualitative data on end user experience.* Thematic analysis will be conducted on qualitative insights derived from FGDs to explore participant experiences, usability, and acceptability of the portal. The analysis will follow the six-phase approach to thematic analysis, which involves familiarization with the data, generating initial codes, searching for themes, reviewing themes, defining and naming themes, and producing the final report [18].

*Cost analysis.* Operational costs and other expenses related to the portal, such as server maintenance, technical support, and staff training, will be tracked for cost analysis. This detailed tracking will enable a comprehensive assessment of the financial feasibility and sustainability of scaling up the platform.

**Sample size estimation.** The primary outcome will be a change in patient activation measure (PAM) [11]. Lin and co-authors found an average effect size of 0.33 in a meta-analysis on studies involving patients with chronic disease reporting change in PAM [19]. Also, a difference of 4 points on the PAM scale is considered clinically important and Marshal et al were able to demonstrate that a 5-point difference was significantly associated with higher odds of CD4 count>200, retention, and viral load suppression [20,21]. Following the definitions as outlined in Hooper et al., 2016 [22], assuming an intra- cluster correlation of 0.02 and individual autocorrelation of 0.8, and only low "time by cluster" effects impacting the PAM scores, the study will require a minimum sample size of 27 per cluster to demonstrate a change of 5 point (green lines in Fig 5) in the activation score between the two groups, assuming the same standard deviation as in Marshall et al, 2013 [21] with a power of at least 90% at an alpha of 0.05% [23].

It is worth noting that the power to demonstrate a difference will be affected substantially by the cluster by time effect. Hemming et al. provided minimal guidance–but did mention a range on that correlation between 0.3 (large effect) to 0.9 (small effect). They strongly suggest a sensitivity analysis on these assumptions, which is provided in Fig 5. Focusing on the green lines, in the presence of high intra-participant variability (individual autocorrelation, IAC) but small time by cluster effects (cluster autocorrelation, CAC), approximately 90% power can be attained with 22 participants per cluster. However, if the time by cluster effect is stronger, power to demonstrate 5 points change drops to 80%. A 10% increase in sample size will be added to account for non-participation, requiring a total of 30 participants per cluster. Based on the above assumptions, the total minimum sample size is therefore estimated at 480. The number of participants lost to follow-up will be monitored throughout the study, and mitigation strategies implemented to accommodate any unforeseen impact by additional recruitment of patients. To compute the sample size for each site, a weighting based on the total AYP will be used. The following formula will be used to obtain a sample for each of the study sites.

$$n_i = \frac{N_i}{N}$$

where $n_i$ = Sample size for site $i$, $N_i$ = total AYP in site $i$ *and* $N$ = total AYP in all the study sites

## Ethical considerations

The AMREF Health Africa ESRC and the respective County health departments have sanctioned this protocol. Every participant involved provided their informed and written consent. For participants aged between 15 to 17 years, the study investigators were granted a waiver for parental consent by the ESRC. This decision was based on several considerations: to avoid

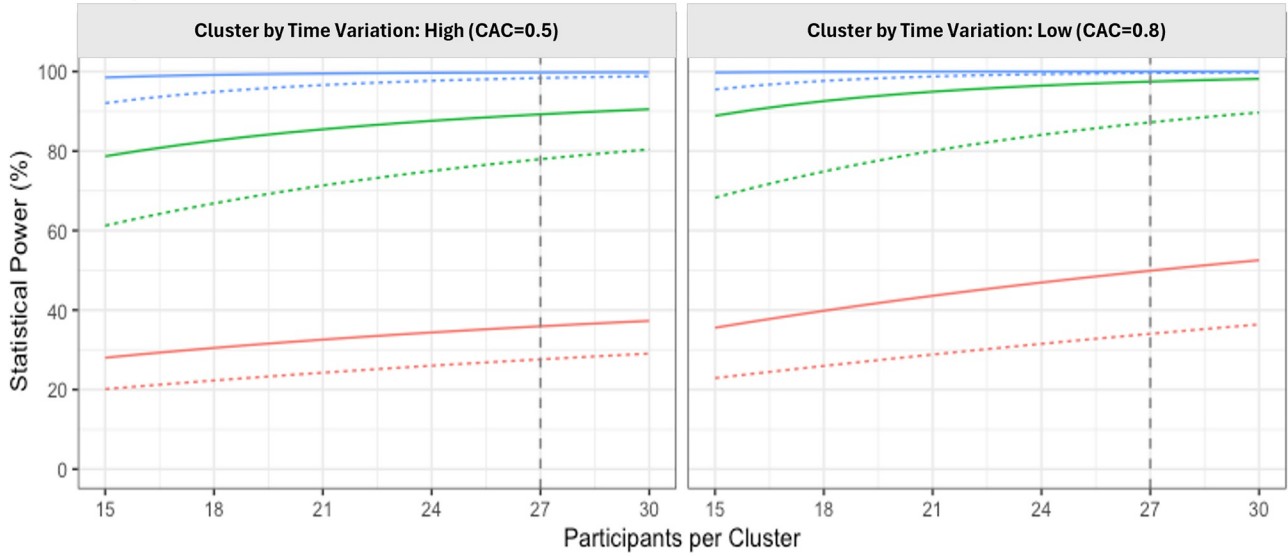

**Fig 5. Sample size estimation for stepped wedge design.**

unintentional revelation of the HIV status, to enhance the participation rate, and to mitigate selection bias, especially since the study entailed minimal risk to its participants. Additionally, the National commission for Science, Technology, and Innovation (NACOSTI) has provided a research license permitting the execution of the study. Furthermore, this trial has been catalogued in the Pan African Clinical Trial Registry at www.pactr.org (ID: PACTR202303729957231).

**Participant consent and data privacy protection.** To safeguard the privacy and data of our participants, we have implemented several critical measures. First, all study investigators have completed comprehensive online training focused on the ethical protection of human subjects in research. Additionally, every member of the study team with access to participant data will be required to sign strict confidentiality agreements before the study begins.

Ensuring participant privacy is a top priority. Interviews will be carefully conducted behind closed doors in designated rooms within the participating facilities, while focus group discussions (FGDs) will take place in outdoor spaces, strategically located within the facility premises to maintain distance from non-participants. Recruitment into the study will only be finalized after obtaining written informed consent, with face-to-face interviews conducted following this crucial step. We will clearly communicate the potential risks and benefits of participation, reinforcing that involvement in the study is entirely voluntary. Written consent for audio recordings will also be obtained from all FGD participants.

For participants under 18 years, identified as 'mature minors,' the informed consent process allows them to consent independently. A waiver of parental consent has been secured to

protect their privacy, particularly for adolescents who may have chosen not to disclose their HIV status to their parents.

To further protect participant anonymity, personal identifiers such as names and addresses have been deliberately excluded from all study data collection tools. Instead, we have used only initials and unique study codes. Informed consent and assent forms, along with any hard copy data collection tools, will be securely stored in lockable, restricted-access areas at the enrolling sites. All electronic data capture and storage tools will be fortified with password protection, and data will be meticulously de-identified to ensure the anonymity of our participants.

## Results

This study will assess the impact of myCareHub on patient activation, medication adherence, social engagement and viral load suppression among AYP living with HIV. We anticipate improvements in these areas:

1. Patient activation—we expect that regular interaction with health diaries, reminders, and screening tools through the myCareHub platform will lead to increased PAM scores in the intervention group, indicating higher levels of patient engagement and activation in managing their health.

2. Social engagement—increased use of online communities through the platform is expected to boost SSQ6 social engagement scores, particularly among those who may face barriers to in-person social interactions.

3. Adherence and viral load suppression—with regular notifications and health management tools, we expect higher medication adherence rates and improved viral suppression among participants who use the portal consistently compared to those in the control group.

## Discussion

The anticipated results highlight the potential of myCareHub to enhance self-management among young people living with HIV. Increased PAM scores would suggest that the portal fosters greater patient activation, while improvements in adherence and viral suppression would confirm its role in supporting clinical outcomes.

Several limitations of the study must be considered. First, selection bias may occur, as participants who are more comfortable with technology might be more likely to engage with the portal, potentially skewing the results. Additionally, confounding factors such as socioeconomic status, access to technology, and health literacy could influence the outcomes. While adjustments will be made during analysis to control for these variables, their impact cannot be entirely eliminated. Another challenge is attrition, as some participants may drop out over time, particularly those with limited access to devices or internet connectivity, which could affect the generalizability of the findings. Technological barriers such as unreliable internet access, compatibility issues with devices, and power outages may also hinder full use of the platform, leading to lower-than-expected engagement and, consequently, a reduced impact on patient outcomes. Finally, the study's generalizability is limited by its focus on specific counties in Kenya, which may restrict the broader applicability of the findings to other regions or populations.

## Conclusion

This study will provide valuable insights into the use of digital health tools like myCareHub to improve patient activation and health outcomes among young people living with HIV. The

results, though promising, must be interpreted within the context of the study's limitations. If successful, myCareHub could serve as a blueprint for expanding digital health interventions to other populations and health conditions, offering a scalable solution to improve healthcare engagement.

## Acknowledgments

We extend our sincere thanks to the County Departments of Health of Kiambu and Kirinyaga Counties for their collaborative support during the study duration. Their cooperation and commitment played a crucial role in the successful execution of this research, providing invaluable local insights and facilitation that ensured the study's relevance and impact.

## Author Contributions

**Conceptualization:** Eric Nturibi, Jared Mecha, Albert Orwa, Justus Kilonzi, Randeep Gill, Sanne Roels.

**Data curation:** Eric Nturibi, Albert Orwa.

**Formal analysis:** Eric Nturibi, Albert Orwa.

**Funding acquisition:** Eric Nturibi, Jared Mecha, Randeep Gill, Sanne Roels.

**Investigation:** Eric Nturibi, Jared Mecha, Elizabeth Kubo, Albert Orwa, Florence Kaara, Faith Musau, Christine Wamuyu.

**Methodology:** Eric Nturibi, Jared Mecha, Albert Orwa, Florence Kaara, Faith Musau, Christine Wamuyu.

**Project administration:** Eric Nturibi, Jared Mecha, Elizabeth Kubo, Florence Kaara, Faith Musau, Christine Wamuyu.

**Resources:** Jared Mecha, Justus Kilonzi, Randeep Gill, Sanne Roels.

**Software:** Albert Orwa, Justus Kilonzi.

**Supervision:** Eric Nturibi, Jared Mecha, Elizabeth Kubo, Albert Orwa, Florence Kaara, Faith Musau, Christine Wamuyu.

**Validation:** Eric Nturibi, Elizabeth Kubo, Albert Orwa.

**Visualization:** Eric Nturibi, Elizabeth Kubo.

**Writing – original draft:** Eric Nturibi.

**Writing – review & editing:** Eric Nturibi, Jared Mecha, Randeep Gill.

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
