## [Decision Letter · Decision Letter 0]

10 Jul 2024

PDIG-D-23-00348

Study protocol for a multi-centre stepped-wedge cluster randomised trial to explore the usability and outcomes among young people living with HIV in Kiambu and Kirinyaga counties of Kenya, using an online health portal

PLOS Digital Health

Dear Dr. Nturibi,

Thank you for submitting your manuscript to PLOS Digital Health. After careful consideration, we feel that it has merit but does not fully meet PLOS Digital Health's publication criteria as it currently stands. Therefore, we invite you to submit a revised version of the manuscript that addresses the points raised during the review process.

Please submit your revised manuscript within 60 days Sep 08 2024 11:59PM. If you will need more time than this to complete your revisions, please reply to this message or contact the journal office at digitalhealth@plos.org. Please include the following items when submitting your revised manuscript:

We look forward to receiving your revised manuscript.

Kind regards,

Dukyong Yoon

Section Editor

PLOS Digital Health

Journal Requirements:

Additional Editor Comments (if provided):

Reviewers' comments:

Reviewer's Responses to Questions

**Comments to the Author**

1. Does this manuscript meet PLOS Digital Health’s publication criteria? Is the manuscript technically sound, and do the data support the conclusions? The manuscript must describe methodologically and ethically rigorous research with conclusions that are appropriately drawn based on the data presented.

Reviewer #1: Yes

Reviewer #2: Yes

2. Has the statistical analysis been performed appropriately and rigorously?

Reviewer #1: Yes

Reviewer #2: Yes

3. Have the authors made all data underlying the findings in their manuscript fully available (please refer to the Data Availability Statement at the start of the manuscript PDF file)?

Reviewer #1: Yes

Reviewer #2: Yes

4. Is the manuscript presented in an intelligible fashion and written in standard English?

Reviewer #1: Yes

Reviewer #2: Yes

5. Review Comments to the Author

Reviewer #1: The manuscript is well-structured and the research design is robust. The use of a stepped-wedge cluster randomized trial is appropriate for this study and is well-executed. Additionally, the statistical methods are appropriately chosen and rigorously applied. Generalized linear mixed models are suitable for handling the complexity of the data.

However, I would like to provide several comments to make this paper clearer and more meaningful.

1. Study Design and Methodology

- While the study design as a stepped wedge cluster randomized trial is well-conceived, there needs to be a more detailed explanation of the characteristics and baseline conditions of each cluster. Specifically, please provide clear strategies and methods for minimizing potential heterogeneity between clusters. 

- In Figure 3, what does the green color represent in addition to the blue and orange colors? Please add an explanation. Also, there is no period after "B". Please ensure consistency with other sections.

2. Data Analysis and Statistical Validity

- The choice of generalized linear mixed models (GLMM) for data analysis is appropriate, but there is a lack of detailed explanation regarding the model specifications and assumptions. Please elaborate on the variable selection process, interaction terms, and handling of potential confounding variables in the modeling process.

3. Portal Usability Evaluation and Implementation

The evaluation of the portal’s usability and implementation is limited. A detailed description of the tools and methods used for usability assessment, the procedures for collecting and incorporating user feedback, and concrete usage data of the portal’s features should be included.

- Especially for the tools used, please specify the original source and whether you obtained approval from the original authors. Additionally, clarify the reliability of these tools.

4. Ethical Considerations

- Although the study mentions ethical approval, there is a lack of detailed explanation regarding participant consent procedures and data privacy protection measures. Given the sensitive nature of health information, please provide clear and specific measures and procedures to ensure data protection and participant privacy.

5. Results Interpretation and Conclusion Drawing

- The interpretation of study results and the drawing of conclusions do not demonstrate the actual impact of portal use on patient activation and secondary outcomes. Please discuss potential confounding factors and limitations in more depth and provide additional analyses and evidence to more convincingly demonstrate the portal’s effectiveness.

- Please provide the full-term explanations for the abbreviated words within the table.

Reviewer #2: Summary 

- While there is a general trend of declining HIV infections across most age groups, the incidence among adolescents and young persons (AYP) in Kenya is on the rise. They have higher technical proficiency compared to other age groups, and therefore have greater access to the online resources. This study analyzed the impact of an online health management portal on the health care of HIV-positive AYPs in Kenya using a multicentre stepped wedge cluster randomized trial. The primary outcome is patient activation, while the secondary outcomes include viral load, social engagement measures, adherence, and portal usage features.

Advantages

- This study offers a fresh perspective by accurately identifying the characteristics of AYPs and analyzing the impact of an online health portal on these patients. 

- Adopting a variety of outcomes made the analysis more useful and comprehensive 

- By determining the sample size with consideration of statistical power, the reliability of study was enhanced 

Concerns

- The readability of Figure 1 is low, making it difficult to discern the relationship between mechanisms and outcomes.

- There are unnecessarily separated sections. For instance, it would be more appropriate to include the "Justification of the stepped wedge design" in the section where the multicentre stepped wedge cluster randomized trial is first introduced

- The intent of the sentence "PAM total scores of 0 and 100 will be considered as missing data, and PAM changes will be analyzed over time and across PAM levels as per the developer’s recommendations" is unclear. It is ambiguous whether it means that missing data will be analyzed over time and across PAM levels as per the developer’s recommendations, or if it separately presents the criteria for missing data and the analysis method for PAM score changes. If it is the latter, additional details on the handling of missing data are required.

- There is a need for adequate explanations of abbreviations included in the tables and figures. For example, "S/N o." in Table 4 and "CAC" in Fig 5 are not mentioned in the text. All abbreviations should be fully spelled out at their first occurrence.

6. PLOS authors have the option to publish the peer review history of their article (what does this mean?). If published, this will include your full peer review and any attached files.

**Do you want your identity to be public for this peer review?** For information about this choice, including consent withdrawal, please see our Privacy Policy.

Reviewer #1: No

Reviewer #2: Yes: Sunyong Yoo

---

## [Decision Letter · Decision Letter 1]

4 Dec 2024

Study protocol for a multi-centre stepped-wedge cluster randomised trial to explore the usability and outcomes among young people living with HIV in Kiambu and Kirinyaga counties of Kenya, using an online health portal

PDIG-D-23-00348R1

Dear Dr Nturibi,

We are pleased to inform you that your manuscript 'Study protocol for a multi-centre stepped-wedge cluster randomised trial to explore the usability and outcomes among young people living with HIV in Kiambu and Kirinyaga counties of Kenya, using an online health portal' has been provisionally accepted for publication in PLOS Digital Health.

Best regards,

Dukyong Yoon

Section Editor

PLOS Digital Health

**Additional Editor Comments (if provided):**

**Reviewer Comments (if any, and for reference):**

Reviewer's Responses to Questions

**Comments to the Author**

1. If the authors have adequately addressed your comments raised in a previous round of review and you feel that this manuscript is now acceptable for publication, you may indicate that here to bypass the “Comments to the Author” section, enter your conflict of interest statement in the “Confidential to Editor” section, and submit your "Accept" recommendation.

Reviewer #2: All comments have been addressed

2. Does this manuscript meet PLOS Digital Health’s publication criteria? Is the manuscript technically sound, and do the data support the conclusions? The manuscript must describe methodologically and ethically rigorous research with conclusions that are appropriately drawn based on the data presented.

Reviewer #2: Yes

3. Has the statistical analysis been performed appropriately and rigorously?

Reviewer #2: Yes

4. Have the authors made all data underlying the findings in their manuscript fully available (please refer to the Data Availability Statement at the start of the manuscript PDF file)?

Reviewer #2: Yes

5. Is the manuscript presented in an intelligible fashion and written in standard English?

Reviewer #2: Yes

6. Review Comments to the Author

Reviewer #2: the authors have taken in to considerations all my suggestions

7. PLOS authors have the option to publish the peer review history of their article (what does this mean?). If published, this will include your full peer review and any attached files.

**Do you want your identity to be public for this peer review?** For information about this choice, including consent withdrawal, please see our Privacy Policy.

Reviewer #2: No
